# Technology Transfer Offices and Their Role with Information Mechanisms for Innovation Performance in Firms: The Case of Ghana

Abdul-Fatahi Abdulai [1], Lyndon Murphy [2], Andrew Thomas [2,*] and Brychan Thomas [3]

1 Faculty of Business, Tamale Technical University, Tamale P.O. Box 3 E/R, Ghana
2 Aberystwyth Business School, Aberystwyth University, Penglais Campus Wales, Aberystwyth, Wales SY23 3DY, UK
3 South Wales Business School, University of South Wales, Pontypridd, Wales CF37 1DL, UK
* Correspondence: ant42@aber.ac.uk

**Abstract:** Research into formal and informal technology transfer between universities and industry in economical developed counties is well-documented. However, such studies are limited in number in developing economies. In the context of developing economies, this study analyses technology transfer offices' role in university technology transfer to Ghanaian firms. We incorporate informal mechanisms as a moderating variable to explore the role of human interaction in the technology transfer value chain. In a cross-sectional survey in Ghana, using structural equation modelling with 245 firms, our research finds a negative moderating effect of informal mechanisms on the effect of technology transfer offices on innovation performance in firms. The findings are of significance to universities and corporate bodies in economically developing nations such as Ghana. Policies to improve the effect of informal mechanisms of university technology transfer offices are proposed in developing economies.

**Keywords:** technology transfer; innovation; business performance; structural equation modelling

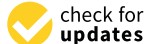



## 1. Introduction

There are many claims of positive impact of innovation in academic literature; nonetheless, researchers still struggle to explain their understanding of how innovation performance in firms is affected by technology transfer offices (TTOs) in developing economies. Neither has the extant literature been able to clearly resolve the debate on the role of informal mechanisms on firms' innovation performance even in developed economies. Moreover, given the volume of innovation literature of late, the absence of a widely agreed theoretical framework to explain the issue is a case for concern, presenting a compelling reason for further investigation into the concept of innovation and influencing factors, and further conceptual and applied research is needed in the discipline of innovation at firm level, and that includes the function of TTOs and informal mechanisms [1].

The global market is now an international technology-based economy given the complexities in the nature of universal market demand and customers' awareness of market dynamics locally and internationally [2]. Following this development, the need for university research and technology, its transfer to industry, and commercialization for economic and social development has become imperative for firms' competitiveness and profitability [3,4]. The days when universities were "Ivory Towers" are over since the establishment of the "third mission" of universities within the last quarter of the twentieth century as a quick response to market demand for technology. This has been exacerbated by the public quest for a direct return on investment in universities [5,6]. In another development, Caniels and Bosch [7] described universities at the time as "cathedrals in the desert"; this was the period when "blue-sky research" dictated the pace of university research, and

knowledge creation was only for peer-reviewed journals. Nowadays, universities and industry are required to operate in a "triple helix" space, where the two team up with governments to generate technology to encourage innovation in firms to ultimately impact society [8–10].

Indeed, available academic literature on technology generation in universities for commercialization purposes is unduly skewed on its incidence in advanced economies [11,12]. The skewed literature delves much into several technology transfer mechanisms, including collaboration, contract research, and joint research, leaving very little space on the more significant intermediary agents of technology dissemination such as TTOs and informal mechanisms [13,14]. In the context of this research, we concern ourselves with technology transfer from universities to firms and investigate the firms' perspective and how informal mechanisms largely determine the level of TTOs influence on innovation performance in firms incrementally at a basic level. As a matter of necessity, the aim of this study is to contribute to the development of literature on the influence of university technology transfer offices on firm-level innovation in developing economies and to include how informal mechanisms of transfer affect the functioning of TTOs in manifesting innovation in firms. To achieve this, we examined the relationship between TTOs and innovation performance in firms in an incremental dimension and the moderating effect of informal mechanisms of transfer. The next section provides a critical review of the literature, followed by the research methodology, where the techniques employed in the research are explained to include the research model and statistical results. The penultimate section is a discussion of the results, with a conclusion in the fifth and final section, which describes the limitations of the research, recommendations for policy reforms, and further research.

## 2. Literature Review

### 2.1. Innovation and Firm's Performance

The meaning of innovation remains a debatable issue among scholars, and the unsettled state of its conceptual definition poses a heavy empirical challenge to researchers. Calling for innovation measurement to reflect on specific industries, sectors', and markets' needs, innovation is an elusive, dynamic, and broad concept that is difficult to define considering the nature of its activities, further demonstrating that deciding on how to measure innovation is a very challenging task to all firms and researchers [15]. The OECD [16] presents a comprehensive definition of innovation:

> An innovation is the implementation of a new or significantly improved product (good or service), or process, a new marketing method, or a new organisational method in business practices, workplace organisation or external relations [16].

One of the simplest definitions refers to innovation as an invention that has been commercialised [17]. Such a definition arguably excludes other forms of innovation, such as organisational innovation [18] and social innovation [19]. In this study, innovation is interpreted as the changes in a firm's operations that lead to the introduction of a novel or improved product, service or process. Indeed, change is considered to be a key element of innovation [20]. Innovation often occurs in a milieux of interconnected individuals, firms, and institutions interacting in feedback loops, promoting innovative activity for business development and economic growth [21]. The cyclical nature of such interactions typically involves institutions, universities, firms, and government agencies. Innovation system theory explains the way in which the generation and transmission of knowledge and technology may support a firm's innovation performance [22]. It is important to note that many factors affect a firm's overall performance; for example, its social media usage [23] and HRM practices [24] may have an impact.

Arguably, this still does not provide clear guidance on how small-scale firms and developing economies can adopt innovation and its models for increased performance in efficiency [25,26]. Notably, at the firm level, innovation measurement captures predominantly any substantial increase in research and development (R&D) budget and other investments in patents, licensing, and spin-offs as achieved from university research and

TTOs [16,27]. A limiting factor found in the literature is the subjective nature of what innovation actually is to different firms and researchers in different disciplines and geographic areas. In reality, what may constitute innovation in one firm may not even be recorded or remembered as such in another firm by some researchers during data collection. Notably, whereas entrepreneurs develop new combinations of existing resources, they remain the drivers of economic development with innovation. Indeed, empirical evidence in many studies shows that innovation and, for that matter, R&D enhance the performance of firms in several respects, including an increase in market share and competitive advantage [28,29]. Although [30] also agreed with this, they contended that the impact of innovation performance is actually weak at industry level.

Innovation is identified as a contributing factor determining a firms' performance, and [31] viewed the impact of innovation from two perspectives: the first being product differentiation or in the introduction of new processes that consolidate a firm's competitive position against its rivals. He, however, warned that profits and growth may not be steady and will usually be temporary or may not last for long if innovating firms cannot continually improve on their core competence and guard against possible replication by competitors. The second perspective, in his view, is the ability of firms to enhance their internal capabilities to respond appropriately and, of course, quickly to market demand compared to their rival firms. Many organizational strategies place emphasis on innovative resources that are internal to firms as the most important drivers of their profitability and strategic advantage [32,33]. This strategic shift drew the attention of management scholars and economic development policy makers towards a resource-based view (RBV) strategy to achieve innovation and economic advantage at firm, regional, and national levels. On the whole, the fundamental question of RBV addresses the issue on why firms are different in deploying their resources and how they achieve and sustain competitive advantage [34,35]. Certainly, the difference can be accounted for in the deployment of innovative approaches in the use of resources and access to university-initiated technology. It is based on this that this study focuses on the technology transfer mechanisms available for firms' innovation performance to offer a direction for evidenced policy and practice.

### 2.2. Technology Transfer Offices

Commercialization of intellectual property through TTOs has received much attention in practice and in the literature [36,37]. For this reason, much has now been seen of such bureaus at local and international levels [30], where they engage university scientists in their research and assess market potential for their findings and breakthroughs [38,39]. Essentially, administrators of TTOs search for prospective investors for a variety of agreements and licensing for university research outputs [2,40]. In fact, scholars still debate the use of the terms technology transfer and technology exchange between universities and industry, and while some are of them consider that technology exchange represents the process better, perhaps due to its multi-directional transmissions of technology between actors, the term technology transfer rather continues to dominate the literature [33,41–43]. Others even prefer technology sharing because of the "public good" characteristics of technology, innovation, and knowledge [44–46]. However, one could argue that there are some similarities with slight differences. Notably, while all the terms represent some form of dissemination of technology, technology transfer portrays a unidirectional transmission from universities to firms, which is the direction of inquiry in this study. Within industry, it is considered to be the transfer of useful know-how or technology across company lines, and while researchers differ in their understanding and definition, they have not yet explained how the transfer happens at industrial sector levels. Additionally, research has not looked deeply into TTOs' innovation activities of economies that have weak innovation systems to offer any understanding, and our knowledge is still limited in that area.

For technology transfer offices in, for example, the U.S. and France, the concept has promoted successful research projects and yielded high levels of income for both universities and investors, breakthroughs, and inventions in university settings. Principally, this

applies to government-funded research [47,48]. Indeed, all of these have been efforts to-wards encouraging universities' effective involvement in innovative activities with industry to influence firms' competence and boost regional and national innovation systems [49,50]. Regrettably, the literature on TTOs suggests that these attempts by policy makers have faced many challenges, primarily in the sense that most university faculty members with break-throughs still use the "backdoor" to get their inventions to market due to bureaucracies and deficiencies in the TTOs' functionalities [2,6,51]. Implicitly, this is due to a lack of proper structures to streamline or supervise the work of university scientists. Typical examples of these abound in developing economies [52,53], where there are no guidelines or a proper general administrative structure to pursue the objectives of university TTOs [54]. A test case is the failure of the Technology Consultancy Centre (TCC) at the Kwame Nkrumah University of Science and Technology in Kumasi in Ghana. Even with that, activities of TTOs in developing economies are vague in the literature [47–49], and for these reasons, this study tries to, by way of empirical study, investigate and contribute knowledge on the potential effect of TTOs on innovation performance in firms in the neglected areas of some world economies such as Ghana. Whilst most available literature acknowledges the significance of TTOs in university technology transfer [55–60], it is virtually silent on how they influence innovation in developing countries such as Ghana. Consequently, it becomes imperative for researchers to expand our understanding on TTOs' activities in those countries and even extend to how different sectors are affected by their work. A more specific study of how interpersonal relationships and informal mechanisms of university technology transfer affect TTOs and their role in innovation still needs to be considered for our understanding of the existing body of technology.

### 2.3. Informal Mechanisms of Transfer

The informal means of university technology transfer is the transfer of technology through relationships between university researchers and individual entrepreneurs, mostly facilitated by human interactions [61]. Known for its extensive tacit characteristics in its mode of diffusion and high intensity of relationships, means of university technology transfer encompass all freely disseminated university sources of technology [54,55]. For instance, an informal communication agreement to offer advice or technical assistance, in the area of technology, is considered informal and requires personal interaction between a university researcher and a co-actor [62,63].

On the whole, informal technology transfer primarily encompasses all non-contractual interactions between technology players and generators and thus university scientists and industry practitioners [61]. It is challenging to differentiate between formal and informal technology transfer in the literature since they are complementary and mutually well-reinforced in an exclusive manner. For example, understood to be a mechanism facilitating the flow of technology through informal communication processes, informal technology transfer comprises consulting, collaborative research, or even technical assistance offered to a firm to achieve efficiency and innovation [64]. On the other hand, a formal technology transfer mechanism is aimed at transferring a well-defined research outcome and with a capitalization intent, such as a patent or a license, which an informal mechanism of tech-nology transfer does not, and there is usually not the slightest expectation to achieve such. Indeed, whereas formal technology transfer ensures a judicious allocation of intellectual property, informal technology transfer does not [61]. The difficulty in the differentiation is compounded even more where research indicates that even if university inventions are publicly disclosed informally, some firms will try to contact scientists and arrange formally to work with them directly [65]. As a result, many university scientists in the U.S. deliberately refuse to disclose their inventions to their universities, as found by [2,66], although they are supposed to do so by law. In a more specific case, [67] raised concerns that informal means of university technology transfer cannot be tracked for any detailed study to add to the body of technology generation literature [64].

Irrespective of the considerable interest in university technology transfer, research on its informal mechanism of transfer in the developing countries' context is quite scant and relatively neglected by researchers given its significance to all firms across the globe. This shows a significant literature deficiency, and this study takes the opportunity to contribute to alleviating this.

*2.4. Conceptual Framework*

Firm-level innovation frameworks, designed to illustrate the innovation process, date back to the 1950s, when scholars attempted to explain the process of innovation in industrial firms [11,68]. As a result, during the post-war years, scientists depicted the process of technology transfer and innovation as one that is smooth, sequential, and in a linear process from start to finish. Namely, the popular "technology push" models [43,69] offered simple and discrete procedural steps to explain how technology is transferred from universities to firms in the industrial world. However, of late, further development in innovation literature indicates that the linear model looked too simple to capture the intricacies of technology transfer in such a straightforward process. Innovation is neither smooth nor linear, nor is it often well-behaved [70]. Indeed, the innovation models of those days were found not to represent the process and therefore considered to be seriously flawed. For instance, the fact that they did not take into consideration the needs of the market presented a clear case. Another criticism levelled against the linear technology transfer models was their concentration on just investment in science, which was only a "technology-push innovation" model in its orientation [17,71]. Still, many critics believe that many facets are overlooked in the linear models. For these models, technology is solely R&D-driven or supply-driven. The major theoretical setback has been the claim that "what is in the market is a product of R&D outcome", with little element of market or customer centralism [11,43]. Above all, the linear models also assume a "one-size-fits-all" status, and a standard criterion has been availed to all practitioners for all forms and categories of technology transfer and innovation processes in the models.

Today, the university technology transfer system is in transition, moving from the marketing and sales of intellectual property model to one that places emphasis on innovation and entrepreneurial university capacity building. In response, scholars such as [72,73] have suggested a model of TTOs development along two dimensions, namely focus and stages of development where legal, marketing, and innovation occupy the focus dimension, as seen in Figure 1, and capacity building and high performance take second stage. The figure shows how various tasks and stages are meticulously executed and how they connect with one another in a systematic nature.

The transmission to an innovation model assumes the status of a broader administrative unit that has business development, entrepreneurial education, and business formation sections. As seen in Figure 1, TTOs have the duty to obtain legal protection for commercial prospects when university invention meets the criteria for exploitation in the market. They go "the extra mile" to shop for users even beyond the expectations of university scientists and negotiate for product licenses, offer advice on R&D, and even go further to form and activate regional networking among technology actors for regional economic development. For all of these to take place, there are roles for informal channels and interpersonal relations that lubricate the process and present social contracts that facilitate technology transfer to firms. Based on the above and advocated for by other sections of TTOs literature [74,75], the study proposes a structural model in Figure 2 and derives three hypotheses to investigate the potential relationship between TTOs and innovation performance in firms. The model also presents a proposition on the effect of informal mechanisms of university technology transfer in ensuring the existence and successful transfer of technology from universities to firms. These hypotheses are as follows:

**H1**: *Informal mechanisms of technology transfer directly influence innovation performance in firms in Ghana.*

**H2**: *Technology transfer offices in Ghana directly influence innovation performance in firms.*

**H3**: *Informal mechanisms of technology transfer directly moderate the relationship between technology transfer offices and innovation performance in firms in Ghana.*

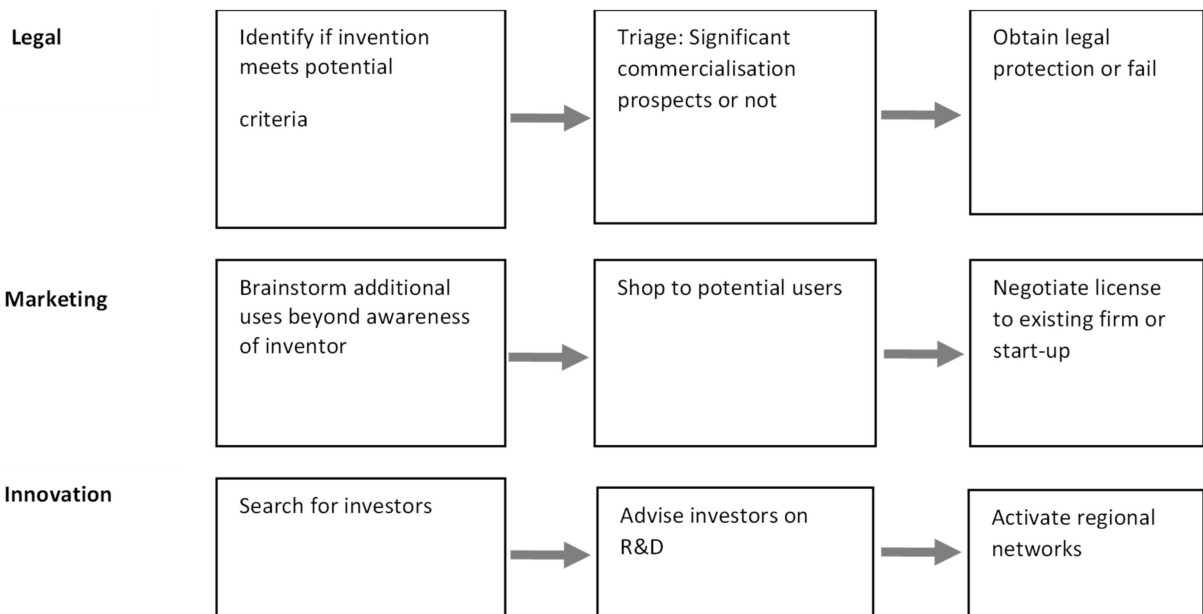

**Figure 1.** Non-linear TTOs stages. Source: [74,75] Etzkowitz and Goktepe-Hulten (2016).

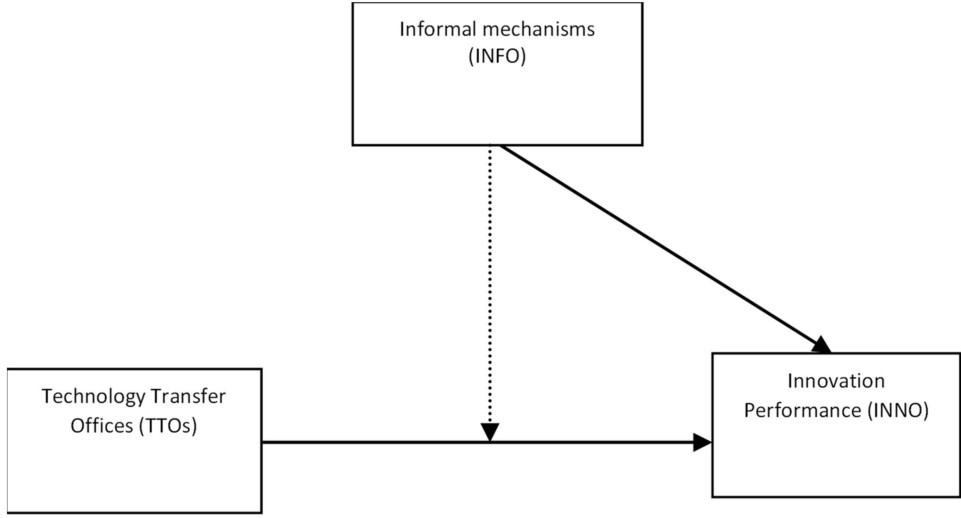

**Figure 2.** Conceptual model. Source: Authors.

### 3. Research Design

To achieve the aim of the research, this study is designed to use a structural equation modelling technique in a hypothesis test study to examine the relationship between the study variables. To do this, a cross-sectional survey questionnaire was constructed and distributed via email to 350 companies in Ghana. Companies were identified from standard databases that contained company names and brief company details that allowed the research team to identify in simple terms whether the company would be suitable for the study. As a result, 245 usable questionnaire returns were obtained and data collected using a stratified simple random sampling method to achieve a fair representative sample of data [76–78]. The questionnaire was designed to obtain information based on the

experiences and knowledge of respondents on forms of informal relationships their firms had and their engagement with TTOs in universities for technology transfer [79].

To measure innovation performance in incremental dimensions in the instrument, both soft and tangible elements were considered to ensure validity of the research findings. Responses on innovation included incremental changes made with technology in products and processes and also firms' budgetary allocation for R&D towards technology acquisitions in a space of one year. Incremental changes made with technology in marketing strategies, general newness in methods in every area of the firms, capacity building, and management style were all factored in. For informal mechanisms, relationships with at least a university academic, including TTOs administrators, was included as well as management interest in published academic literature and common association with university research staff at personal levels. Finally, with TTOs, firms' involvement with TTOs and their staff for the purpose of technological innovation, breakthroughs, and possible spin-offs were rated by respondents.

All firms in the survey were privately owned and included wholesale, retail, and processing firms; ICT organizations; and technology delivery firms. In fact, two databases from the Association of Ghana Industries (AGI) and National Small-Scale Industries (NB-SSI), now the National Enterprise Foundation, were accessed to obtain a sample frame of 800 firms. The three major industry sectors were included, and after two follow-up requests, a 30.63% response rate was obtained [80]. For a medium effect size of 0.10, a sample size of 100 was estimated a priori for a power of 0.80 [80]. A partial least square (PLS)–structural equation modelling (SEM) algorithm with WarpPLS v5.0 statistical software was used [81].

PLS-SEM is a component-based, second-generation multivariate data analysis technique with a capacity to model multiple exogenous and endogenous latent variables in a single structural model [81–83]. The technique comes with many advantages over its counterparts, e.g., ordinary least squares. Missing data were managed with multiple regression imputation [84], and the measurement models were reflectively measured [85]. A bootstrapping algorithm was used to estimate the model parameters and standard errors with 5000 resamples in five iterations [81,86,87]. Figure 3 presents statistical results of the PLS technique, giving the path coefficients and their significant test results.

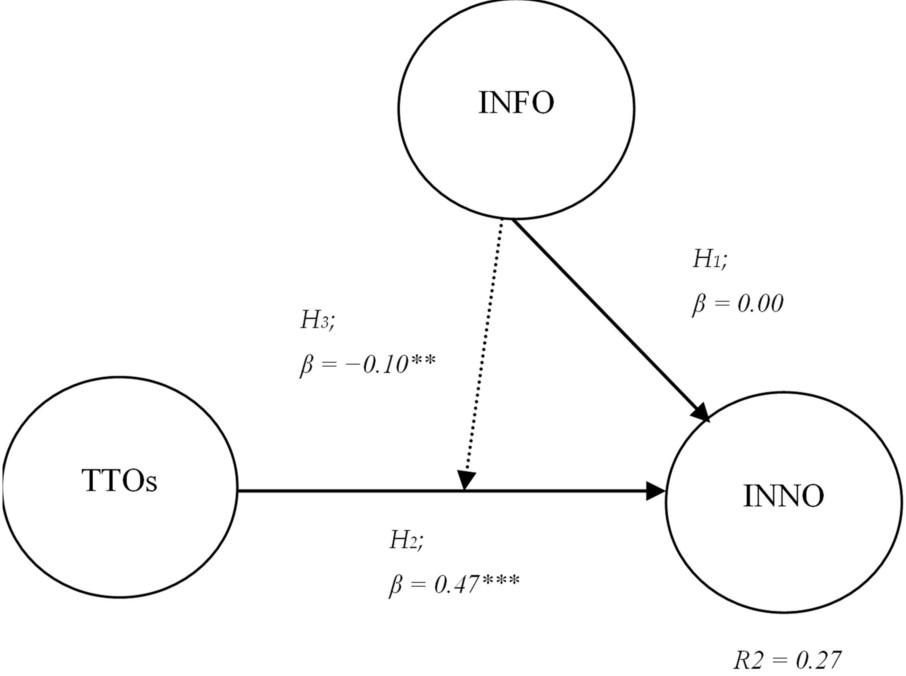

**Figure 3.** Structural model. Source: Authors. *** $p < 0.05$; ** $p < 0.001$.

*3.1. Exploratory Data Analysis*

Table 1 gives demographic information of the sample and shows that of the 245 firms, 36.73% were within 10 to 25 years of age and were the highest representation of all the categories in Table 1. The least category was 12.65%, with firms between 26 and 50 years of age. However, start-ups and other younger firms aged between 1 and 9 years were 30.20%, whilst firms above 50 years were 18.78%. Firms fifty years of age and over were 18.78%, giving a rough distribution of firms' operations in the last half a century in the country. At the least, the data capture a comprehensive section of the business community in Ghana and guarantee a valid generalization of the study findings [88].

**Table 1.** Firms age distribution, frequencies, and percentages (N = 245). Source: Author data.

| Age Range | Frequency | Percentage (%) | Cumulative (%) |
|-----------|-----------|----------------|----------------|
| 1–9 | 74 | 30.20 | 30.20 |
| 10–25 | 90 | 36.73 | 66.93 |
| 26–50 | 31 | 12.65 | 79.58 |
| 50+ | 46 | 18.78 | 98.36 |
| Not Declared | 4 | 1.64 | 100.00 |
| Total | 245 | 100.00 | |

*3.2. Model Evaluation*

The measurement models, which were reflectively measured in the structural models, were evaluated for fitness to the data in terms of the construct's internal consistency and reliability. Factor loading and Cronbach alpha ($\alpha$) values were within acceptable thresholds and can be seen in Table 2 [83]; thus, INNO is 0.742, INFO is 0.764, TTOs is 0.801, and INFO+ is 0.923, respectively. For convergent validity between latent variables and their observable indicators, the highest outer loadings obtained are also consistent with a recommended threshold of 0.50 or higher [83,89]. Discriminant validity was achieved and indicated by the square roots of the average variance extracted (AVE), as seen in Table 3, as it was assessed to be greater than each variable's highest correlation coefficient with other variables [90,91], and VIF reported are less than 3.3, showing no multicollinearity problem [92].

Finally, uni-dimensionality and convergent validity are also achieved, where the manifest variables can be seen to have converged and loaded rightly at more than 0.60 only on their respective constructs [86]. In Table 4, where indices for evaluating the structural models can be found, the coefficients of determination ($R^2$s) reveal the explanatory powers of the exogenous latent variables, which shows a low coefficient (0.001) for $H_1$, an average effect of 0.244 for $H_2$, and an effect of 0.028 for $H_3$, which also indicates a low effect. Similarly, effect sizes ($f^2$) are found to be in good range for all relationships, with $Q^2$s being above zero ($Q^2 > 0.00$): a proof of the relevance of the two exogenous variables on the endogenous variable [93].

**Table 2.** Combined loadings of the measurement models. Source: Author data.

| Variables | INNO | INFO | TTOs | INFO + | SE | *p*-Value |
|---|---|---|---|---|---|---|
| Inn17 | 0.659 | −0.069 | −0.083 | −0.093 | 0.057 | <0.001 |
| Inn18 | 0.739 | 0.021 | 0.082 | −0.032 | 0.056 | <0.001 |
| Inn19 | 0.786 | −0.048 | 0.119 | 0.071 | 0.056 | <0.001 |
| Inn20 | 0.645 | 0.048 | −0.231 | 0.068 | 0.057 | <0.001 |
| Inn21 | 0.678 | 0.054 | 0.073 | −0.022 | 0.057 | <0.001 |
| Inf6 | 0.052 | 0.755 | −0.068 | −0.008 | 0.056 | <0.001 |
| Inf7 | 0.089 | 0.720 | −0.190 | 0.085 | 0.056 | <0.001 |
| Inf8 | 0.072 | 0.707 | 0.069 | −0.061 | 0.057 | <0.001 |
| Inf9 | −0.042 | 0.736 | 0.067 | −0.011 | 0.056 | <0.001 |
| Inf10 | −0.184 | 0.670 | 0.135 | −0.005 | 0.057 | <0.001 |
| Ttr12 | −0.138 | 0.268 | 0.638 | 0.150 | 0.057 | <0.001 |
| Ttr13 | 0.008 | 0.022 | 0.757 | −0.006 | 0.056 | <0.001 |
| Ttr14 | −0.114 | −0.078 | 0.783 | −0.068 | 0.056 | <0.001 |
| Ttr15 | 0.155 | −0.108 | 0.794 | 0.030 | 0.056 | <0.001 |
| Ttr16 | 0.064 | −0.055 | 0.756 | −0.082 | 0.056 | <0.001 |
| Inf6*Ttr | 0.162 | −0.033 | 0.023 | 0.563 | 0.058 | <0.001 |
| Inf6*Ttr | 0.072 | 0.069 | −0.104 | 0.574 | 0.058 | <0.001 |
| Inf6*Ttr | −0.059 | 0.031 | −0.012 | 0.678 | 0.057 | <0.001 |
| Inf6*Ttr | 0.055 | 0.018 | −0.044 | 0.673 | 0.057 | <0.001 |
| Inf6*Ttr | 0.120 | 0.041 | −0.192 | 0.626 | 0.057 | <0.001 |
| Inf7*Ttr | 0.100 | −0.024 | 0.218 | 0.514 | 0.058 | <0.001 |
| Inf7*Ttr | 0.000 | 0.097 | 0.018 | 0.608 | 0.057 | <0.001 |
| Inf7*Ttr | 0.033 | 0.080 | 0.004 | 0.664 | 0.057 | <0.001 |
| Inf7*Ttr | 0.061 | 0.093 | 0.028 | 0.665 | 0.057 | <0.001 |
| Inf7*Ttr | 0.071 | 0.156 | −0.084 | 0.591 | 0.058 | <0.001 |
| Inf8*Ttr | −0.082 | −0.288 | 0.386 | 0.535 | 0.058 | <0.001 |
| Inf8*Ttr | −0.179 | −0.305 | 0.321 | 0.572 | 0.058 | <0.001 |
| Inf8*Ttr | −0.354 | −0.105 | 0.185 | 0.635 | 0.057 | <0.001 |
| Inf8*Ttr | −0.241 | −0.142 | 0.235 | 0.650 | 0.057 | <0.001 |
| Inf8*Ttr | −0.311 | −0.029 | 0.107 | 0.608 | 0.057 | <0.001 |
| Inf9*Ttr | 0.112 | −0.317 | 0.339 | 0.504 | 0.059 | <0.001 |
| Inf9*Ttr | −0.136 | −0.214 | 0.239 | 0.530 | 0.058 | <0.001 |
| Inf9*Ttr | −0.217 | −0.031 | 0.162 | 0.630 | 0.057 | <0.001 |
| Inf9*Ttr | −0.199 | −0.042 | 0.205 | 0.630 | 0.057 | <0.001 |
| Inf9*Ttr | −0.154 | 0.014 | 0.026 | 0.587 | 0.058 | <0.001 |
| Inf10*Ttr | 0.386 | −0.036 | −0.171 | 0.497 | 0.059 | <0.001 |
| Inf10*Ttr | 0.402 | 0.134 | −0.451 | 0.544 | 0.058 | <0.001 |
| Inf10*Ttr | 0.165 | 0.278 | −0.527 | 0.545 | 0.058 | <0.001 |
| Inf10*Ttr | 0.210 | 0.173 | −0.374 | 0.602 | 0.058 | <0.001 |
| Inf10*Ttr | 0.167 | 0.296 | −0.545 | 0.579 | 0.058 | <0.001 |
| *α* | *0.742* | *0.764* | *0.801* | *0.923* | | |

+ Represents moderating variables in the model. * Represents interaction between the two variables as inf is seen to be moderating the influence of Ttr on Inno.

**Table 3.** Correlations among latent variables and square root of AVEs. Source: Author data.

| Variables | (1) | (2) | (3) | (4) VIF |
|---|---|---|---|---|
| (1) INNO | 0.703 * | | | |
| (2) INFO | 0.226 *** | | 0.718<br>1.23 | |
| (3) TTOs | 0.510 *** | 0.457 *** | 0.748<br>1.72 | |
| (4) INNFO | −0.214 *** | −0.128 ** | −0.202 *** | 0.594<br>1.54 |

Note: Square root of average variance extracted (AVEs) shown on diagonal sign. *, *p* < 0.10; **, *p* < 0.05; ***, *p* < 0.001.

**Table 4.** Structural model validation. Source: Author data.

| | Description | Threshold | Path (Hypothesis) | Values Achieved | Outcome |
|---|---|---|---|---|---|
| $R^2$ | This is a measure of the variance explained by the exogenous latent variable of the total variance in the endogenous. | Substantial = 0.670, average = 0.333, and as low = 0.190 [82] | H1 | 0.001 | This explains virtually nothing of the variation in the endogenous latent variable (no power). |
| | $(0 \geq R^2 \leq 1)$ | Substantial = 0.75, moderate = 0.50, and weak = 0.25 [83] | H2 | 0.244 | This explains a relatively about average variation of the total variation in the endogenous latent variable (average effect). |
| | | | H3 | 0.028 | This explains a relatively low variation of the total variation in the endogenous latent variable (low effect). |
| $f^2$ | This measures the impact of the exogenous latent variable on the endogenous latent variable. $(0 \geq f^2 \leq 1)$ | Low = 0.020, medium = 0.150, and large = 0.350 [80] | H1 | 0.001 | This is a relatively low impact of the exogenous latent variable on the endogenous latent variable (no effect). |
| | | | H2 | 0.244 | This is a low impact of the exogenous latent variable on the endogenous latent variable (large effect). |
| | | | H3 | 0.028 | This is a low impact of the exogenous latent variable on the endogenous latent variable (low effect). |
| $Q^2$ | This measures the predictive relevance of the endogenous latent variable to the endogenous latent variable. | [81] | H1 | 0.276 | This appears as a relevant exogenous latent variable to the endogenous latent variable (relevant). |
| | | | H2 | 0.276 | |
| | $Q^2 > 0.00$ | | H3 | NA | |

Note: $R^2 = 0.27$ is the combined contribution of two latent variables as seen in Figure 2.

*3.3. Model Results*

Table 5 presents the structural model results from Figure 2 and shows a non-significant relationship ($H_1$: $\beta = 0.004$, $p > 0.05$) between INFO and INNO. However, TTOs is statistically significant and with a positive effect ($H_2$: $\beta = 0.477$, $p < 0.05$) on INNO, while INFO is statistically significant with a negative effect ($H_3$: $\beta = -0.106$, $p < 0.05$) on the relationship between TTOs and INNO as a moderating variable.

**Table 5.** Path coefficients for latent variables. Source: Author data.

| Path (Hypothesis) | Direct Effect ($\beta$) | SE | $p$-Value | Moderation Effect ($\beta$) | SE | $p$-Value |
|---|---|---|---|---|---|---|
| $H_1$ | 0.004 | 0.064 | 0.475 | NA | - | - |
| $H_2$ | 0.477 | 0.059 | 0.001 | NA | - | - |
| $H_3$ | NA | - | - | −0.106 | 0.063 | 0.047 |

## 4. Discussion

In summary, the data analysis reveals the role of informal mechanisms of university technology transfer and TTOs in terms of their influence on innovation found in firms in developing economies, using Ghana as a test case. The results of our hypotheses tests (shown in Table 5 and described in Section 3.3) show that for H1, the results are not confirmed, whilst H2 is confirmed, and H3 is also confirmed. Furthermore, the hypotheses tests showed that TTOs, which are mostly referred to as technology or innovation centres in Ghana, positively affect innovation in firms and lead to up to 47% of innovation in Ghanaian firms. Implicitly, TTOs play a significant role in facilitating the marketisation of university research breakthroughs. Evidence of this includes spin-offs and investment in start-ups, enabling innovation and wealth creation to take place in firms. The findings are demonstrated in the outcome of the work of [94], which sought to investigate the productivity of French TTOs after government reforms. Significantly, their study revealed that 50% of improvements in TTOs' productivity reflected in the data envelopment analysis (DEA)-based Malmquist productivity index, which was ascribed to the productivity of the French TTOs systems. It further reports that the improvements are the result of both positive efficiency and technology change. Notwithstanding the fact that younger offices with hospitals showed negative productivity, the overall productivity is a demonstration of TTOs' positive effect on commercialization of intellectual property in France. Similar positive results from TTOs were reported by [95] following support funds granted by the Scientific and Technological Research Council of Turkey (TUBITAK). Accordingly, demonstrable results in the form of patents, licenses, and new businesses were reported.

Conversely, informal mechanisms of university technology transfer have shown no direct influence on innovative changes that take place in firms in Ghana, according to our findings. Put another way, social relationships through which most informal technology transfer mechanisms are conducted and those associated with networks and social capital have no material impact on improvement of firms' products and services.

Additionally, informal mechanisms of university technology transfer negatively affect the direct influence of TTOs on innovation performance in the sense that as more firms are drawn into using informal mechanisms of university technology transfer, the involvement of TTOs in technology transfer from universities to firms reduces. Consequently, this is found to constitute a disincentive to the efforts and activities of TTOs in Ghana. Indeed, for every unit increase in informal mechanisms of university technology transfer in Ghana, the study found a corresponding 10% decrease in TTOs' influence on innovation performance or, better put, a reduction in industry players' engagement in university spin-offs and start-ups. This may demonstrate the level of frustration TTOs face in places and markets where firms have low absorptive capacities due to a low level of education attainment and lack of basic functional skills by most business owners and firms' staff. Arguably, with higher informal means used, TTOs' involvement in technology transfer, patent, and licensing rather leads to loss of investment and spin-offs and start-ups. This eventually renders production of technology in universities virtually worthless, thereby keeping a great amount of research on shelves in universities. Interestingly, in a study by [96] in China, informal mechanisms of technology transfer, which they explained to revolve around trust, were found to greatly facilitate technology flows in exchange relationships in emerging markets, which is actually at variance with the findings in this study. Nonetheless, trust and control were found to jointly affect technology transfer negatively, as found in this study, where informal mechanisms hinder the relationship between TTOs and innovation in firms. It could be stated that informal mechanisms of university technology transfer do not support progress in the wake of mistrust and dishonesty, and when a society is embroiled with doubts and suspicions, technology transfer of any form suffers. This assertion is evident in Table 2, where the manifest variable that measures trust as a significant component in social capital loaded negative as well as Inf7 ($-0.190$), among others, with TTOs latent variable and even as a moderating variable and inf7*Trt ($-0.24$) as well. This view is supported by [89], who concluded with a similar finding, calling it "a

dark side" of social capital, and it may be understood why such a valuable means to free technology does not support innovation performance in firms in Ghana.

The research outputs from this study provide contributions to knowledge and decision making in that the role of TTOs clearly provides added value to the development of innovation in firms, whereas informal mechanisms of knowledge and technology transfer have little effect, and in some cases, they impede the innovation and knowledge transfer process in companies largely due to incorrect information being provided, thus leading to incorrect decisions being made within companies. This further identifies the influence of institutions such as TTOs in positively affecting the decision-making processes within companies through providing clear and sound information to firms. Therefore, it is imperative that firms in Ghana utilize the capacity and capabilities of TTOs to assist in their development so that innovation can thrive within the engaged businesses.

## 5. Conclusions

The issues of the influence of informal mechanisms of university technology transfer and TTOs have been found to have different effects on innovation performance in firms, according to the study findings. Of course, the two means of university technology transfer will therefore need different responses from stakeholders to understand and address the subjects pertaining to their significance in the national innovation system of Ghana. Primarily, the insignificance of informal means of technology transfer in getting university research and discoveries to firms for innovation requires a joint effort of innovation management professionals and universities to work toward recognizing the value and role they can play in fostering innovation. Indeed, the free nature of some channels for university technology and easy access of university researchers as social change factors present a great deal of potential for firms. Particularly, small and financially weak firms require and depend on informal mechanisms for university technology, as claimed by [89,90], to gain and achieve innovation and to increase productivity. Without doubt, inventions are tedious and costly, and no firms should believe they can benefit fully from them without contributing their fair share of the cost.

Nonetheless, the study found TTOs to demonstrate an effective and significant role in innovation through their successes in linking investors and existing businesses with universities for bringing breakthroughs to the market in Ghana. This finding could be taken advantage of by universities to encourage and support their TTOs, as done by the Technology Venture Corporation (TVC) in New Mexico [53], to gain more income to augment their internally generated fund (IGF) with income from industry.

Industry players can also create value through patent and licensing from TTOs' activities with intellectual property, which could be profitable locally and internationally. A recent decline in government research budgets across the world due to economic crises sends a signal to universities and researchers to look elsewhere for extra income [96–100]. Technology transfer offices could play a significant role, as was found in the study; in that sense, by fostering closer and stronger links between university researchers and industry, higher levels of income could be gained from intellectual property generated in universities. The government of Ghana could boost TTOs' chances of success with policies to support in-service training and further capacity building for TTOs' staff. This could be undertaken in collaboration with higher-ranking TTOs and universities at both local and international levels. Incentives could be made available by the Government of Ghana for universities as a policy by adopting a modified version of the Bayh-Dole Act of 1980 by the U.S. government [48,101]. Furthermore, tax holidays could be given to industries patronizing TTOs' services, and public procurement of products' regulations could be directed at goods and services produced from such links [101].

Policies need to be designed to institute functional literacy and education for less-educated business owners and workshops and seminars to give confidence and increase the absorptive capacities of less-educated entrepreneurs in Ghana. This is believed to be able to strengthen their links with universities and bring expected results as found in more

economically developed countries. It is believed that other countries in a similar situation could benefit from lessons learnt from these findings and recommendations. The study was limited to the ten administrative areas of the country, which could affect the generalizability. However, some urban centres were captured for data in an effort to achieve valid results. We therefore recommend further research that should capture the entire country of Ghana, achieving data more representative of the country.

**Author Contributions:** Conceptualization, A.-F.A. and L.M.; methodology, A.-F.A.; software, A.-F.A.; validation, B.T. and A.T.; formal analysis, A.-F.A.; investigation, L.M.; resources, B.T.; data curation, B.T.; writing—original draft preparation, A.-F.A.; writing—review and editing, L.M., B.T. and A.T.; visualization, A.T.; supervision, L.M.; project administration, L.M.; funding acquisition, N/A. All authors have read and agreed to the published version of the manuscript.

**Funding:** This research received no external funding.

**Institutional Review Board Statement:** Not applicable.

**Informed Consent Statement:** Informed consent was obtained from all subjects involved in the study.

**Data Availability:** Data for this study is available upon request.

**Conflicts of Interest:** The authors declare no conflict of interest.

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
