# Peer review of "Technology Transfer Offices and Their Role with Information Mechanisms for Innovation Performance in Firms: The Case of Ghana"

_knowledge, doi:10.3390/knowledge2040041_

Round 1

Author Response

Thank you for your suggestions to improve the paper. The manuscript has been fully updated with specific consideration given to the following:

Reviewer 1

Introduction:

In the introduction, although the author (s) puts forward the survey data to prove the importance of the study, it only provides support for the research object, without any basis to support the importance and necessity of the research content. The remaining part is not clear about the importance and significance of the research.

Answer: the paper is updated to show the contribution to the special edition theme

Theory and Hypotheses:

In this part, the author (s) did not mention any theory that underpins this research! Does this study have a theory that underpins this research? In addition, in regard to the hypotheses, although this structure can be used, the logical structure of the writing is too messy for me to understand clearly. Moreover, the hypothesis setting is also not clear in this paper. Thus, the contribution is quite weak due to the unclear mechanism.

Answer: The paper has been re-written to provide additional information and restructuring to make it clearer to the reviewer

This paper lacks theoretical support, which makes me uncertain about the reliability of the research model. Particularly, there is a fair amount of literature on this issue. Not all relevant literature in this domain is covered in this study and some of the literature that is covered could be discussed with more nuance (e.g., there is a range of studies on recent with field samples, thus it is not correct to say that the majority of the studies only focuses on the lab). Please, these references may be useful to include in this study:

Answer: We have included some of the above references in the paper. Some of the references suggested were not suitable in all cases as they did not match the theme of the paper.

In the part of the discussion and conclusion, the author (s) spent a lot of time sorting out the research findings and practical implications. The description of the part of theoretical implications is more like the research conclusions. I do not find the theoretical contribution of the research to the existing research field. I think it is necessary for the author (s) to add and highlight the theoretical implications, and the discussion of the results should be as simple and clear as possible. So, please also make sure to articulate your contribution (or how your paper advances knowledge in this field) both in the Introduction and in the Conclusion as well as the novelty of your paper and implications for future research. Kindly also make sure to highlight the practical implications of your research from the perspective of organizations and managers, as well as policymakers in more depth.

Answer: The paper has been re-written to provide additional information and restructuring to make it clearer to the reviewer

Methodology & Data Analysis:

Further explanation about, the data collection section needs to address the data collection technique/s, how the data were collected, what methods and mechanisms were used to obtain the data, how was the sample selected, how representative the sample is if it is.

Answer: Further detail added to the research design section

B- Importantly, how did the author (s) treat the CMV? Before and after the data collection? (Have the author (s) used any remedies?

Reviewer 2 Report

The paper is well structured, well written, clearly and consistently explains the methodology used and the results are well presented. There is space for improvement mainly in three components, such as: 1) how the reflection presented enriches one or more domains of the journal's special issue call - what is the specific contribution of the paper to the reflection on knowledge and decision-making processes; 2) the vast majority of the references are very dated, it would be advisable to update the literature review with more current references and 3) there is no real discussion of the results; the results are presented but not very well discussed and crossed with the theoretical references.

More detailed comments are included in the review file.

Author Response

Thank you for your suggestions to improve the paper. The manuscript has been fully updated with specific consideration given to the following:

Reviewer 2

  • how the reflection presented enriches one or more domains of the journal's special issue call - what is the specific contribution of the paper to the reflection on knowledge and decision-making processes;

Answer: Discussions ection updated with controbution toteh journal SI themes

  • the vast majority of the references are very dated, it would be advisable to update the literature review with more current references and

Answer: Additional, more current references added throughout

  • there is no real discussion of the results; the results are presented but not very well discussed and crossed with the theoretical references.

Answer: further details added to the discussion section to connect with theoretical themes

Reviewer 3 Report

Dear Authors,

The article is clear, well written and systematized.

  In my opinion the article "Technology Transfer Offices and Their Role with Information Mechanisms for Innovation Performance in Firms: The Case of Ghana" is "Accepted after minor revision" for publication in Journal Knowledge. There are, however, small suggestions that can be taken into consideration (attached document).

Best regards

Author Response

Thank you for your suggestions to improve the paper. The manuscript has been fully updated with specific consideration given to the following:

Reviewer 3

Research Design
What constitutes a moderating effect in statistics could be added.

Answer: moderating effect details added in section 3,4
Discussion
The integration of confirmation/non-confirmation of each of the hypotheses is missing:
Answer: confirmation / non-confirmation of hypotheses Included in discussion
References
References are well numbered but not written in accordance with the instructions for authors.
References should be described as follows, depending on the type of work:

Answer: All references checked and updated

Round 2

Reviewer 2 Report

The revised version complies, in general terms, with the improvement proposals made, particularly in the component of better discussion of the results. It appears that the author or authors also included new references, although it was expected that their relevance would be greater, most of the references used are still not very current.
From the point of view of text formatting, the source of the tables and graphics prepared by the author/authors is still not indicated, as well as, in table 2, starting from the line with the variable "Inf6*Ttr", the formatting of the table needs to be corrected.

Author Response

Thank you for your suggestions and requirements for improvement of the paper. The paper has been revised in line with your comments. Updates include:

  1. References updated throughout the paper to remove any pre 2000 dated papers plus updates on more relevant and up to date references added
  2. Alignment of table 2 data

Reviewer 3 Report

Dear Authors,

Some formal aspects should be taken into account:

1) From line 87 to line 99: the spaces between lines are different;

2) In table 2, starting from the variable Inf6* Ttr, the values of the columns must be realigned;

3) line 412: Instead of "H 2" it should be "H2"

Best regards

Reviewer

Author Response

Dear Reviewer

Thank you for your recommendations for improving the paper. We have fully revised the paper in line with your comments. Adjustments made are:

1) From line 87 to line 99: the spaces between lines are different;

Respaced to ensure continuation with remainder of document

2) In table 2, starting from the variable Inf6* Ttr, the values of the columns must be realigned;

Realigned to match the reamining items in the table

3) line 412: Instead of "H 2" it should be "H2

Retyped to H2

Thank you